# Alkaloids and Colon Cancer: Molecular Mechanisms and Therapeutic Implications for Cell Cycle Arrest

**DOI:** 10.3390/molecules27030920

**Published:** 2022-01-28

**Authors:** Haroon Khan, Waqas Alam, Khalaf F. Alsharif, Michael Aschner, Samreen Pervez, Luciano Saso

**Affiliations:** 1Department of Pharmacy, Abdul Wali Khan University Mardan, Mardan 23200, Pakistan; waqasalamyousafzai@gmail.com; 2Department of Clinical Laboratory, College of Applied Medical Science, Taif University, P.O. Box 11099,Taif 21944, Saudi Arabia; alsharif@tu.edu.sa; 3Department of Molecular Pharmacology, Albert Einstein College of Medicine, Bronx, NY 10461, USA; michael.aschner@einsteinmed.org; 4Department of Pharmacy, Qurtuba University of Science and Information Technology, Peshawar 29050, Pakistan; samreengcp@gmail.com; 5Department of Physiology and Pharmacology “Vittorio Erspamer”, Sapienza University of Rome, 00185 Rome, Italy; luciano.saso@uniroma1.it

**Keywords:** colon cancer, alkaloids, anticancer activity, cell cycle arrest

## Abstract

Cancer is the second most fatal disease worldwide, with colon cancer being the third most prevalent and fatal form of cancer in several Western countries. The risk of acquisition of resistance to chemotherapy remains a significant hurdle in the management of various types of cancer, especially colon cancer. Therefore, it is essential to develop alternative treatment modalities. Naturally occurring alkaloids have been shown to regulate various mechanistic pathways linked to cell proliferation, cell cycle, and metastasis. This review aims to shed light on the potential of alkaloids as anti-colon-cancer chemotherapy agents that can modulate or arrest the cell cycle. Preclinical investigated alkaloids have shown anti-colon cancer activities and inhibition of cancer cell proliferation via cell cycle arrest at different stages, suggesting that alkaloids may have the potential to act as anticancer molecules.

## 1. Introduction

Cancer is the second most fatal disease in the world, and according to the World Health Organization (WHO), it accounted for approximately 8.2 million deaths and 14 million new victims in 2012 [1]. In 2017, the projected deaths due to different cancers in the United States were 600,920, with 1,688,780 newly diagnosed cancer patients. The overall cancer prevalence and death rate was 20% higher in males than in females [2]. The coronavirus disease 2019 (COVID-19) pandemic impeded cancer detection and therapy in 2020. In 2021, the United States was projected to have1,898,160 cancer cases and 608,570 cancer-related deaths [3]. Older people, between 85–90 years-of-age, have higher cancer prevalence than younger subjects [4]. The rapid and unmanageable proliferation of abnormal cells leads to cancer origination [5], with abnormalities during cellular division in the mitochondrial genome triggering genetic mutations and cancer progression [4].

Colon cancer (CC) is one of the most prevalent cancers in the world, with the highest prevalence in Black Alaska’s Natives and the lowest prevalence in Asian/Pacific Islanders [6]. Colorectal cancer (CRC) is the most prevalent cancer in Saudi men, and the third most prevalent disease in Saudi women [7]. The American Cancer Society has drafted guidelines for the treatment of colon cancer as well as survivorship care, which include long-term complications from chemotherapy [8,9,10].

While diagnostic and treatment options for colon cancer have been stipulated [11,12,13], their outcomes are not always satisfactory, and the number of deaths remains high. Colon cancer is the third most prevalent type of cancer, concerning both cases and deaths in the USA [14]. Even though synthetic drugs are currently available, they suffer from low efficacy and adverse side effects, which has led to treatment failure and/or low patient compliance [15,16]. Along with drug treatments, diet and physical exercise may also offer treatment [17,18].

Monoclonal antibodies and small-molecule inhibitors are used in targeted therapeutics and are regarded as a viable therapy alternative. Chemotherapy has been the most prevalent treatment of CC; nevertheless, various limitations, such as limited specificity, inappropriate biodistribution, and poor pharmacokinetic profile, often compromise its effectiveness. As a result, researchers have searched for novel chemotherapeutic drugs that are both highly safe and efficacious [19].

Natural products from medicinal plants have been previously used in the treatment of several diseases [20,21,22,23,24,25,26,27]. Among these natural products, alkaloids are one of the most diverse and extensively investigated classes of compounds [28,29,30,31]. Alkaloids have been shown to possess antiplatelet [32], antioxidant [33,34], antinociceptive [35], antipyretic [36], antiasthmatic [37], anticancer [38], and antibacterial effects [38,39] and to inhibit acetylcholinesterase (AChE) and angiotensin-converting enzyme(ACE) inhibition [40].

In this review, we have summarized and focused on the anticancer potentials of various plant-derived alkaloids against colon cancer that could be translated into clinically beneficial agents with a well-defined mechanism—cell cycle arrest.

## 2. Cancer Treatment—Alkaloids

Extracts and alkaloids isolated from plants have shown a role in the suppression of oncogenesis [41,42]. Alkaloids have demonstrated an effect on the regulation of various mechanistic pathways involved in proliferation, cell cycle, and metastasis, thereby attracting considerable attention [43,44]. From a clinical perspective, vincristine and vinblastine have historical importance as anticancer agents and are clinically used in the treatment of acute lymphoblastic leukemia [45]. Antitumor drugs, such as vincristine, vinblastine, and paclitaxel, which are being utilized in clinical practice, can exist naturally. Additionally, vincristine or vinblastine treatment has been shown to limit cancer growth in the majority of patients with complicated hemangiomas [46]. These botanicals act through multiple mechanisms and have shown efficacy in breast, ovarian, non-small cell lung, and prostate cancers [47,48,49]. Berberine (BBR), a fully natural isoquinoline alkaloid obtained from various botanical groupings, has recently attracted attention. Berberine has been shown to be beneficial for immunotherapy by acting as a dopamine receptor antagonist and suppressing the release of Interferon gamma (IFN-γ), Tumor Necrosis Factor-alpha (TNF-α), Interleukin-1 beta (IL-1β) and Interleukin-6 (IL-6) from LPS-activated cells. BBR has been shown to inhibit tumor proliferation, to induce autophagy and apoptosis, and to suppress metastasis and angiogenesis [50].

## 3. Cell Cycle Arrest in Cancer Treatment

Cell cycle arrest is a dynamic and versatile mechanistic pathway that has been implicated in the pathophysiology of various human systems [51,52], and which is characterized by several cell cycle checkpoints (G1, S, G2, M) [53,54], especially at the transitions from the G1 phase to the S phase, and from the G2 phase to the M phase [55,56].

### 3.1. G1-Phase Arrest

During the G1 phase, cells prepare for entry into the cell cycle, and duplicate their DNA in the S phase [57,58]. The Gap 1 (G1) checkpoint controls the DNA status and cellular activities and controls DNA replication in the synthesis phase (Sphase) [59,60]. The regulatory gene BDE-47 has been shown to cause marked attenuation in cell proliferation targeting the G1 phase [61] by modulating p53 and p21 expression, which in turn regulated cyclinD1 and CDK2. The overall process is influenced by Dux4 stimulation, whichcausesp21 mRNA and protein expression and cell cycle arrest through G1 phase arrest [62]. The S transcription in the G1 phase is strongly associated with the transcription factors of E2F coupled with their dimerization partner proteins. Researchers have found functional regulation factors of E2F in cancer [63,64], suggesting that G1-S transcription is critical in the development and proliferation of cancer [65,66,67].

### 3.2. G2-Phase Arrest

The S phase is followed by the Gap 2 (G2) phase, characterized by a synthesis/repair phase coupled with the preparation of cellular machinery for mitosis in the M (mitosis) phase [68,69]. The chromatids and daughter cells separate during the M phase. Subsequently, the cells enter theG1 or G0. The quality and rate of overall cell division is modulated by these checkpoints [70,71]. In this phase, the cdc2–B1 complex is regulated via the activation of phosphate by the Cdk-activating enzyme and the inhibition of phosphatase enzymes [72]. The phosphatase Cdc25C caused dephosphorylation and hence overexpression of Cdk [73,74,75].

The dormant cells are activated by external stimuli [76]. These growth-promoting factors bind to cell receptors and trigger the cellular machinery [77]. Moreover, genetic intervention has been observed as the focusing target of these growth-modulating factors [78]. The response is designated as earlier and delayed. The former induces phosphorylation and activation of the transcription factor proteins that are already present in the cell. In fact, the genes that are involved in early response encode transcription factors, and ultimately modulate the expression of late response genes. The delayed response genes modulate the proteins, G1 cyclin-dependent kinases (CDKs) [79,80], and therefore CDK inhibitors have recently attracted considerable attention as anticancer agents, especially the CDK4 and CDK6 inhibitors [81,82].

## 4. Activity of Alkaloids against Colon Cancer

In 2001, Ogasawara et al. reported on the anticancer activity of evodiamine **1** (Table 1), a quinolone alkaloid, isolated from the fruits of *Evodia rutaecarpa*, a traditional Chinese medicinal plant. This alkaloid caused a 70% reduction in the formation of lung metastases induced by colon carcinoma 26-L5 cells in mice (10 µg/mL in 48 h). Likewise, Zhao et al. and Huang et al. reported multiple mechanisms of colon-cancer inhibition induced by this alkaloid.

Isostrychnopentamine (ISP)**2** is an indolomonoterpenic alkaloid present in the leaves of *Strychnos usambarensis*, an African shrub. Frédérich et al. (2002) demonstrated the anticancer activity of this alkaloid against HCT-116 and HCT-15 human colon cancer cells. In both cases, ISP caused cell death (IC_50_ = 7.0 µM for HCT-116 and IC_50_ = 15.0 µM for HCT-15), promoting cell cycle arrest in the G2-M phase and inducing different pathways of apoptosis. Four years later, Jiao et al. (2006) isolated and identified chaetominine **3** from *Chaetomium* sp., an endophytic fungus present on the leaves of *Adenophora axilliflora*. Compound **3** was active against SW-116 colon cancer cells with an IC_50_ value of 28.0 nM.

Shoeb et al. (2006) demonstrated the antiproliferative activity of a unique, dimeric indole alkaloid isolated from the seeds of *Centaura montana*, named montamine **4**. This alkaloid showed significant activity against CaCO_2_ cells (IC_50_ = 43.9 µM), whereas its monomer, moschamine **5**, showed an IC_50_ value of 81.0 µM.

Koduru and co-researchers (2007) isolated steroidal alkaloids, named as tomatidine **6** and solasodine **7,** from *Solanum aculeastrum* (Figure 1) [83]. Sanguinarine **8**, a benzophenanthridine alkaloid isolated from the roots of *Sanguinaria canadensis*, evoked a significant anticancer effect [84]. Subsequently, in 2012, Lee et al. reported other work about the antiproliferative activity of this alkaloid [85].

**Table 1 molecules-27-00920-t001:** Plant-derived alkaloids with anticancer effects in colon cancer.

Plant Species	Alkaloids with Structure and Potency	Reference
*Evodia rutaecarpa*	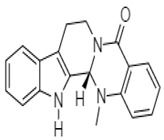 **1** (IC_70_: 10 µg/mL)	[86]
*Strychnos usambarensis*	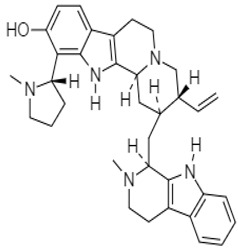 **2** (IC_50_: 7.0–15 µM)	[87]
*Chaetomium* sp.	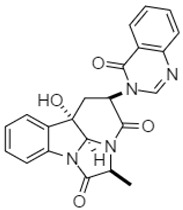 **3** (IC_50_: 28.0 nM)	[88]
*Centauramontana*	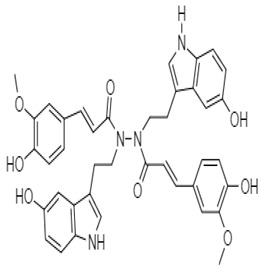 **4** (IC_50_: 43.9 µM)	[89]
	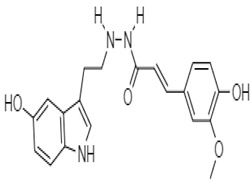 **5** (IC_50_: 81.0 µM)	
*Solanum aculeastrum*	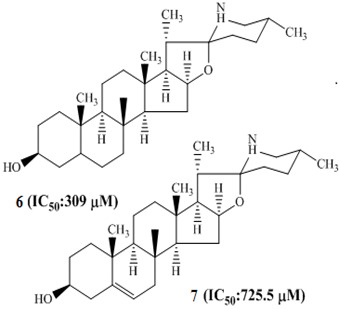	[83]
*Sanguinaria canadensis*	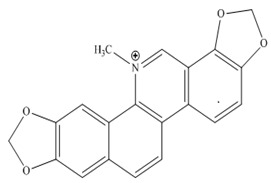 **8** (IC_78.5_: 1 mM)	[84]
*Corydalis ternate*	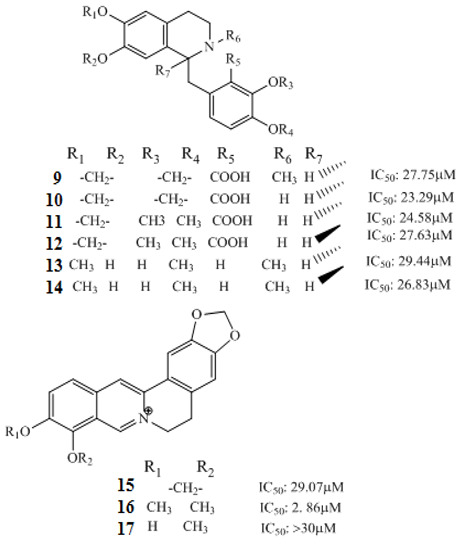 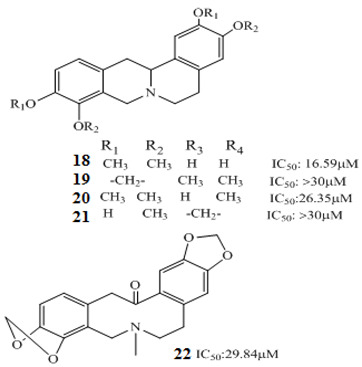	[90]
*Menispermum dauricum*	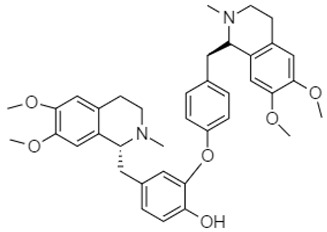 **23** (IC_80_: 10 µM)	[91]
*Evodia rutaecarpa*	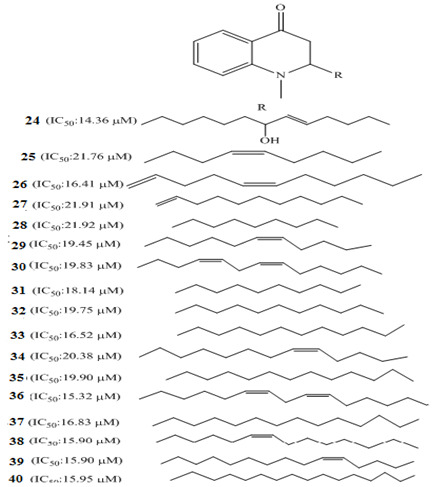	[92]
*Cynanchum paniculatum*	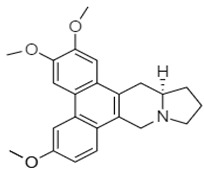 **41** (IC_50_: 4.7–10.8 nM)	[93]
*Zanthoxylum capense*	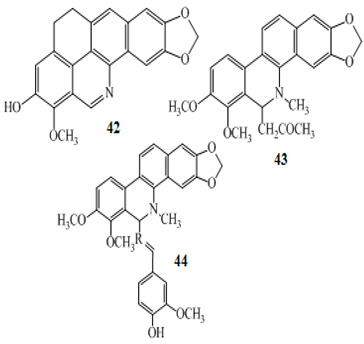	[94]
*Acorus gramineus*	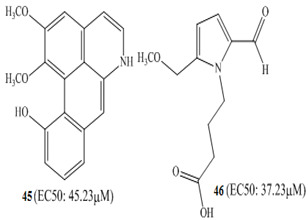	[95]
*Tabernaemontana corymbosa*	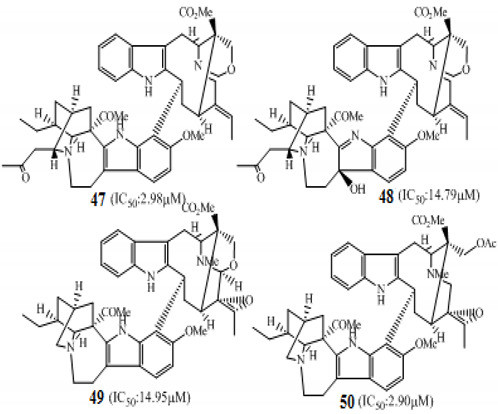	[96]
*Piper nigrum*	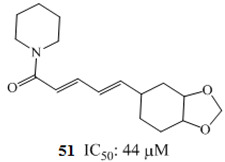	[97,98]
*Tabernaemontana elegans* *vobasinyl−iboga* *Alkaloids*	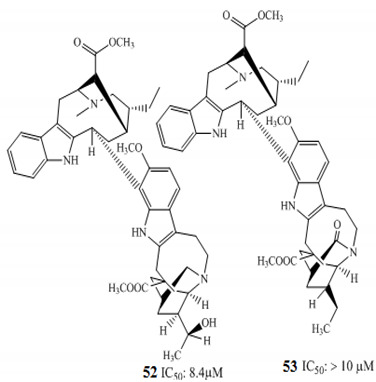	[99]
*Melodinus henryi*	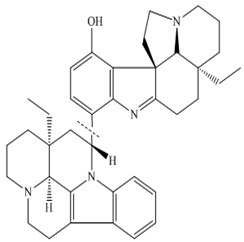 **54** (IC_50_: 4.9 µM)	[100]
*Argemone mexicana*	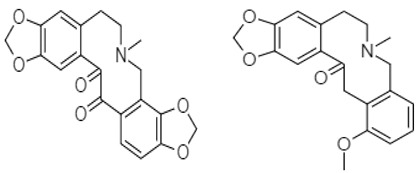 **55** (200 µg/mL: 24–28% reduction) **56** (200 µg/mL: 24–28% reduction) 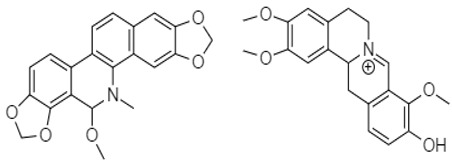 **57** (200 µg/mL: 100% reduction) **58**(200 µg/mL: 48% reduction) 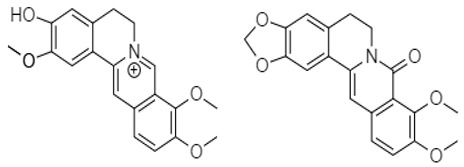 **59** (200 µg/mL: 100% reduction) **60** (200 µg/mL: 78% reduction)	[101]
*Murraya koenigii*	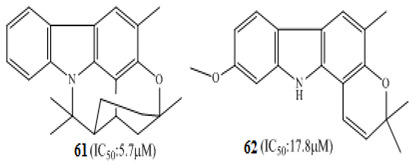	[102]
*Goniothalamus lanceolatus*	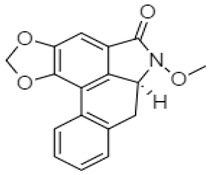 **63** (IC_50_ = 5.32 µM)	[103]

Korean researchers isolated several alkaloids **9–22** from *Corydalis ternate* phytochemicallyy [90]. These were characterized as epi-coryximine **9**, coryternatines A **10**, coryternatines B **11**, coryternatines C **12**, coryternatines D **13**, (*S*)-reticuline, (*R*)-reticuline **14**, coptisine **15**, dimethyl corydalmine **16**, tetrahydroberberine **17**,berberine **18**, protopine **19**, corydalmine **20**, thalifendine **21**, and cheilanthifoline **22**. When studied for in vitro anticancer activity against various cancer cell lines, they showed selective cytotoxicity against human colon cell line, HCT-15.

Yang et al. (2010) demonstrated that dauricine **23** (Table 1), a bisbenzylisoquinoline alkaloid, inhibited four lines of colon cancer cells (HCT-116, HCT-8, SW620, and SW480). The mechanism involved the inhibition of proliferation/invasion and induction of apoptosis by suppressing nuclear factor-kappaB (NF-kB) activation in a dose/time-dependent relation. In 2012, investigation of the whole plant of *Evodia rutaecarpa* by Chinese researchers led to the isolation of 17 quinolone alkaloids. The anticancer studies of these compounds caused marked antiproliferative activity against gastric cell N-87 in vitro assay. These alkaloids are 1-methyl-2-[7-hydroxy-(*E*)-9-tridecenyl]-4(1*H*)-quinolone **24** (IC_50_: 16.25 μM), 1-methyl-2-[(*Z*)-4-nonenyl]-4(1*H*)-quinolone **25** (IC_50_: 17.25 μM), 1-methyl-2-[(1*E*,5*Z*)-1,5-undecadienyl]-4(1*H*)-quinolone **26** (IC_50_: 16.70 μM), 1-methyl-2-[(*E*)-1-undecenyl]-4(1*H*)-quinolone **27** (IC_50_: 18.04 μM), 1-methyl-2-nonyl-4(1*H*)-quinolone **28** (IC_50_: 23.34 μM), 1-methyl-2-[(*Z*)-6-undecenyl-4(1*H*)-quinolone **29** (IC_50_: 18.66 μM), 1-methyl-2-[(4*Z*,7*Z*)-tridecadienyl]-4(1*H*)-quinolone **30** (IC_50_: 17.85 μM), 1-methyl-2-decyl-4(1*H*)-quinolone **31** (IC_50_: 21.69 μM), 1-methyl-2-undecyl-4(1*H*)-quinolone **32** (IC_50_: 20.52 μM), 1-methyl-2-dodecyl-4(1*H*)-quinolone **33** (IC_50_: 20.82 μM), evocarpine **34** (IC_50_: 17.25 μM), dihydroevocarpine **35** (IC_50_: 18.18 μM), 1-methyl-2-[(6*Z*,9*Z*)-pentadecadienyl]-4(1*H*)-quinolone **36** (IC_50_: 12.64 μM), 1-methyl-2-tetradecyl-4(1*H*)-quinolone **37** (IC_50_: 16.72 μM), 1-methyl-2-[(*Z*)-6-pentadecenyl]-4(1*H*)-quinolone **38** (IC_50_: 14.52 μM), 1-methyl-2-[(*Z*)-10-pentadecenyl]-4(1*H*)-quinolone **39** (IC_50_: 14.52 μM), and 1-methyl-2-pentadecyl-4(1*H*)-quinolone **40** (IC_50_: 14.27 μM) [92].

Antofine **41**, a phenanthroindolizidine alkaloid isolated from the whole plant of *Cynanchum paniculatum*, showed potent antiproliferative effects in different types of colon cancer cells with IC_50_ values in the nanomolar range (6.3 nM on HCT-116, 10.8 nM on HT-29 and 4.7 nM on SW-480). In HCT-116 cells, treatment with antofine induced cell cycle arrest in theG0/G1 phases (Min et al., 2012).

Three other alkaloids **42**–**44** were isolated from *Zanthoxylum capense*; when subjected to anticancer activity against HCT-116 colon cancer cells using the Guava ViaCount viability assay at various test concentrations of 5, 10, 20, and 50 Μm, significant effects were observed [94]. These were named after characterization as decarine **42**, norchelerythrine **43** and zanthocapensine **44**. Kim et al. (2015) worked on the isolation of bioactive secondary metabolites from Korean medicinal plants, isolating from *Acorus gramineus* two alkaloids, gramichunosin **45** and 4-(2-Formyl-5-(methoxymethyl)-1*H*-pyrrole-1-yl)butanoic acid **46 [95]**.When these compounds were studied for anticancer potential against the colon cancer cell line HCT-15, both the alkaloids were found with EC_50_: 45.23 and 37.23 µM, respectively.

Zhang and coworkers (2015) isolated four new alkaloids from *Tabernaemontana corymbosa,* named as tabercorines A–C (**47**–**49**) and 17-acetyl-tabernaecorymbosine A **50 [96]**. In the results against SW-480 cell lines, tabercorines A **47** (IC_50_: 9.24 µM) and 17-acetyl-tabernaecorymbosine A **50** (IC_50_: 4.70 µM) were more potent than standard cisplatin (IC_50_: 11.07 µM).Yaffe et al. (2015) isolated piperine **51** from *Piper nigrum,* which exhibited marked anticancer effects against colon cancer cell lines with IC_50_: 44 µM [97]. Researchers from Portugal investigated a monoterpene indole alkaloid isolated from *Tabernaemontana elegans* [99]. Dregamine **52** and tabernaemontanine **53** caused 8.4 µM and >10 µM, respectively. Liu et al. (2016) isolated melodinineV **54** from *Melodinus henryi* [100]. When studied against the HT-29 colon cancer cell line, it showed potent anticancer effects with an IC_50_ value of 4.9 µM in vitro. Singh et al. isolated six different alkaloids (13-oxoprotopine **55**, protomexicine **56**, 8-methoxydihydrosanguinarine **57**, dehydrocorydalmine **58**, jatrorrhizine **59**, and 8-oxyberberine **60**) from the whole plant of *Argemone mexicana* and studied them against the SW-480 human colon cancer cell line. Of these test compounds, jatrorrhizine and 8-methoxydihydrosanguinarine caused 100% cell death after 48 h at 200 µg/mL [101].

Arun et al. (2017) isolated pyranocarbazole alkaloids from the leaves and flowers of *Murraya koenigii,* from which murrayazoline **61** and *O*-methylmurrayamine A **62** showed potent anticancer activity in DLD-1coloncancercells [102], with IC_50_ values of 5.7 mM and 17.9 mM, respectively. An alkaloid isolated from *Goniothalamus lanceolatus*, namely, goniolanceolactam **63**, showed potent cytotoxicity against the colon cell line with IC_50_ value of 5.32 µM [103]. A schematic representation of molecular pathways of the anticancer effect of steroidal alkaloids in colon cancer is presented in Figure 1.

## 5. Activity of Synthetic Derivatives against Colon Cancer

The summary of the activity of synthetic alkaloid derivatives with potential therapeutic effect in colon cancer is shown in Table 2. The derivative exatecan mesylate ((1*S*,9*S*)-1-amino-9-ethyl-5-fluoro-2,3-dihydro-9-hydroxy-4-methyl-1*H*,12*H*-benzo(de) pyrano(3′,4′-6,7)indolizino(1,2-b)quinoline-10,13(9*H*,15*H*)dionemethanesulfonate) **64** from camptothecin, a quinoline indole alkaloid with antiproliferative activity, was investigated by Hattum et al. (2002) for anticancer activity against several colon cancer cell lines (COLO205-IC_50_: 0.61 ng/mL, COLO320-IC_50_: 0.52 ng/mL, LS174T-IC_50_: 2.8 ng/mL, SW-1398-IC_50_: 2.7 ng/mL, and WiDr-IC_50_: 7.5 ng/mL, for in vitro activity). The activity was also observed in an in vivo model, and this derivative was considered more potent than topotecan and SN-38 [104].

In 2007, Italian researchers synthesized nortopsentin analogues of marine alkaloids, such as 3,5-bis(3′-indolyl)pyrazol **65** and 1-chloro-3,5-bis(3′-indolyl)pyrazol **66**, earlierisolated from *Spongosorites* (Table 2). These analogues showed activity against several colon cancer cells (COLO-205: GI_50_ = 7.98 and 2.22 µM, respectively; HCC-2998: 4.74 and 1.71 µM; HCT-116: 19.0 and 3.82 µM; HCT-15: 8.68 and 3.01 µM; HT-29: 5.35 and 3.57 µM; KM-12: 4.58 and 3.45 µM; and SW-620:>100 and 3.52 µM). The analogue 1-chloro-3,5-bis(3′-indolyl) pyrazol, bearing a chloro group, was more active both in terms of GI_50_ and of percentage of sensitive cell lines. Plakinamines N and O, two additional steroidal alkaloids from *Corticium niger*, as well as two established plakinamine compounds, were isolated (3, 4). Using a combination of MS and NMR spectroscopic data, the structures of these molecules were characterized. The antineoplastic effectiveness of plakinamines N, O, and J was examined in the NCI-60 screen, and they demonstrated improved inhibitory activity against all colorectal cell cultures [105].

**Table 2 molecules-27-00920-t002:** Synthetic alkaloid derivatives with potential in colon cancer.

Sources	Derivatives			Reference
exatecan mesylate	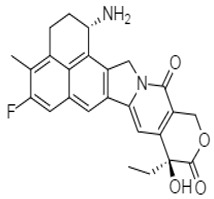 **64** = (IC_50_ = 0.5–7.5 ng/mL)			[104]
Nortopsentin Analogues	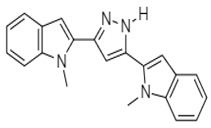 **65** (IC_50_ = 4.58–19.0 µM)			[106]
	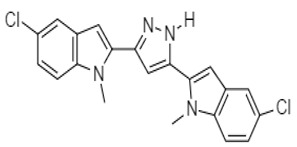 **66** (IC_50_ = 1.71–3.82 µM)			
Harmine analogue	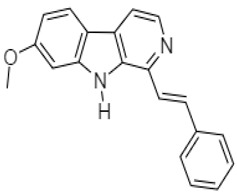 **67** (IC_50_: 6.6–19.0 µM)			[107]
amide alkaloid derivatives	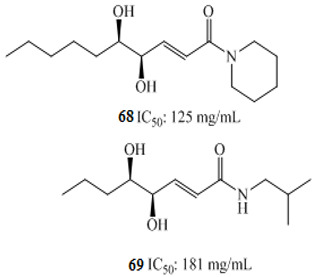			[108]
5-Phenyl-4,5-dihydro-1,3,4-thiadiazole Analogues	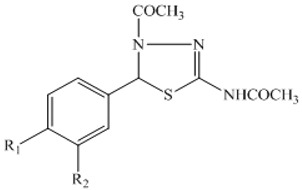			[109]
		R1	R2	
	**70** (IC_50_: 10.25 µg/mL)	H	H	
	**71**(IC_50_: 10.25 µg/mL)	CH_3_	H	
	**72** (IC_50_: 10.25 µg/mL)	NO_2_	H	
	**73** (IC_50_: 10.25 µg/mL)	H	NO_2_	
	**74** (IC_50_: 10.25 µg/mL)	NCH_2_	H	
	**75** (IC_50_: 10.25 µg/mL)	Cl	H	
	**76** (IC_50_: 10.25 µg/mL)	Br	H	
	**77** (IC_50_: 10.25 µg/mL)	OH	H	
	**78** (IC_50_: 10.25 µg/mL)	OCH_3_	H	
	**79** (IC_50_: 10.25 µg/mL)	OH	OCH_3_	
Berberine Chloride	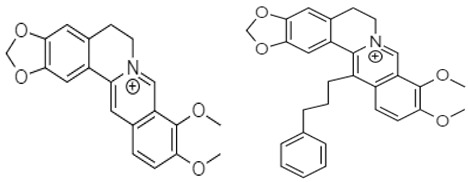 **80** (IC_50_: 31.97–36.63 µM) **81** (IC_50_: 10 µM) 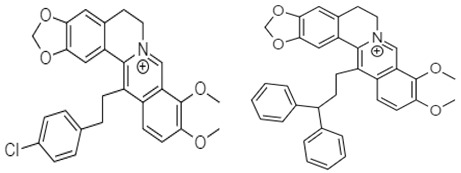 **82** (IC_50_: 10 µM) **83** (IC_70_: 10 µM)			[110]
Berberine chloride	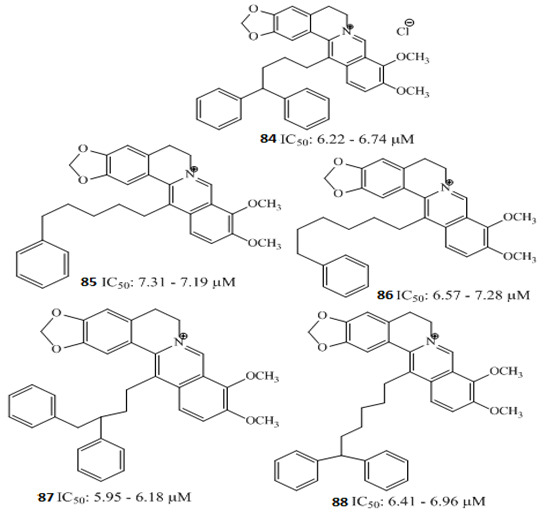			[111]
Dregamine and Tabernaemontamine	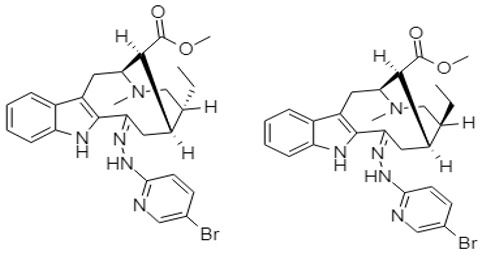 **89** (IC_95_: 25 µM) **90** (IC_95_: 25 µM) 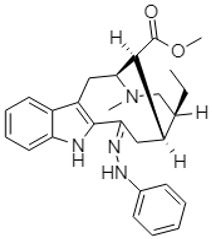 **91** (IC_95_: 50 µM)			[112]
Phenanthroindolizidine	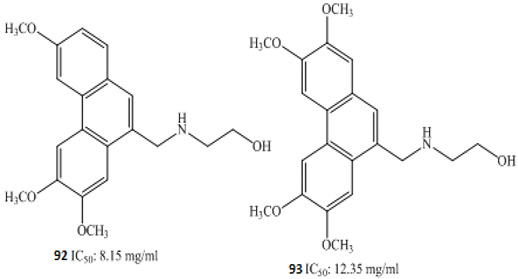			[113]

Luo et al. investigated the anticancer effect of JKA97 (methoxy-1-styryl-9*H*-pyrid-[3,4-*b*]-indole) **67**, a benzylidene analogue of alkaloid harmine. This synthetic substance inhibited human colon cancer HCT-116 cells by apoptotic induction via p53-independent mechanisms. The authors demonstrated that JKA97-induced apoptosis was impaired in Bax knock-out (Bax^−/−^) HCT-116 cells. In addition, JKA97 also showed significant apoptotic activity against another human colon cancer cell line with p53 mutations (SW-620).Srinivas et al. (2009) synthesized amid alkaloids which were characterized as (4*R*,5*R*)-2*E*-4,5-dihydroxy-1-(piperidin-1-yl)dec-2-en-1-one **68** and (4*R*,5*R*)-2*E*-4,5-dihydroxy-*N*-isobutyloct-2-enamide **69 [108]**.

These stereoselective amid alkaloids exhibited a marked anticancer effect against HT-29 colon cell lines with 125 and 181µg/mL, respectively. Similarly, several 5-Phenyl-4,5-dihydro-1,3,4-thiadiazole derivatives were synthesized by Alam and colleagues. These include *N*-(4-Acetyl-5-phenyl-4,5-dihydro-1,3,4-thiadiazol-2-yl)acetamide **70**, *N*-(4-Acetyl-5-p-tolyl-4,5-dihydro-1,3,4-thiadiazol-2-yl)acetamide **71**, *N*-(4-Acetyl-5-(4-nitrophenyl)-4,5-dihydro-1,3,4-thiadiazol-2-yl)-acetamide **72**, *N*-(4-Acetyl-5-(3-nitrophenyl)-4,5-dihydro-1,3,4-thiadiazol-2-yl)-acetamide **73**, *N*-(4-Acetyl-5-(3-(dimethylamino)phenyl)-4,5-dihydro-1,3,4-thiadiazol-2-yl)acetamide **74**, *N*-(4-Acetyl-5-(4-chlorophenyl)-4,5-dihydro-1,3,4-thiadiazol-2-yl)-acetamide **75**, *N*-(4-Acetyl-5-(4-bromophenyl)-4,5-dihydro-1,3,4-thiadiazol-2-yl)-acetamide **76**, *N*-(4-Acetyl-5-(4-hydroxyphenyl)-4,5-dihydro-1,3,4-thiadiazol-2-yl)-acetamide **77**, *N*-(4-Acetyl-5-(4-methoxyphenyl)-4,5-dihydro-1,3,4-thiadiazol-2-yl)-acetamide **78**, and *N*-(4-Acetyl-5-(4-hydroxy-3-methoxyphenyl)-4,5-dihydro-1,3,4-thiadiazol-2-yl)acetamide **79 [109]**.

Berberine **80** is a highly studied isoquinoline plant alkaloid isolated from many species of the genus berberis [114]. Italian researchers synthesized a number of berberine **80** derivatives, which were characterized as [13-(3-phenylpropyl)-9,10-dimethoxy-5,6-dihydrobenzo [g]-1,3-benzojiokisoro [5,6-a] quinolizinium iodide] **81**, [13-[2-(4-chlorophenyl) ethyl]-910-dimethoxy-5,6-dihydrobenzo [g]-1,3-benzojiokisoro [5,6-a] quinolizinium iodide] **82**, [13-(3,3-diphenyl-propyl)-9,10-dimethoxy-5,6-dihydrobenzo[g]-1,3-benzojiokisoro [5, 6-a] quinolizinium iodide] **83**, [13-(4,4-diphenylbutyl)-9,10-dimethoxy-5,6-dihydrobenzo[g]-1,3-benzodioxolo [5,6-a]quinolizinium chloride] **84**, [13-(5-phenylpentyl)-9,10-dimethoxy-5,6-dihydrobenzo[g]-1,3-benzodioxolo[5,6-a]quinolizinium chloride] **85**, [13-(6-phenylhexyl)-9,10-dimethoxy-5,6-dihydrobenzo[g]-1,3-benzodioxolo[5,6-a]quinolizinium chloride] **86**, [13-(5,5-diphenylpentyl)-9,10-dimethoxy-5,6-dihydrobenzo[g]-1,3-benzodioxolo[5,6-a]quinolizinium chloride] **87** and [13-(6,6-diphenylhexyl)-9,10-dimethoxy-5,6-dihydrobenzo[g]-1,3-benzodioxolo[5,6-a]quinolizinium chloride] **88**. Compounds **81**, **82,** and **83** were active against HCT-116 and SW-613-B3 cells. The evaluation of cell survival by a DNA release-based assay revealed that10 µM NAX012 produced an inhibitory effect on HCT-116 cell survival (40%) and enhanced during the recovery time (50%). NAX014 induced a cell growth inhibition of approximately30%, which increased to about 50% with the recovery time, whereas NAX018 demonstrated a cytotoxic effect after the treatment in about 50%, reaching 70% after the recovery time. InSW-613-B3 cells, the incubation with 10 µM NAX012 and NAX014 was not effective, while NAX018 affected cell survived by 10% and 20% (at the completion of incubation and after the recovery response, respectively).In particular, when compared to the lead compound BBR, these derivatives showed high potency, being more effective in cells harboring p53wt, promoting cell cycle arrest and DNA damage, and triggering caspase-dependent apoptosis and autophagy [111]. Ilyas and his coauthors established the anticancer efficacy of doxorubicin (of synthetic origin) combined with berberine (of natural origin)by applying an encapsulation/conjugation approach with poly (lactic-co-glycolic acid) PLGA nanoparticles. By using carbodiimide chemistry, doxorubicin has been effectively conjugated to PLGA, and the PLGA–doxorubicin conjugate (PDC) was employed to encapsulate berberine. ROS analysis demonstrated that the PDBNP depicted a dose-dependent rise in the reactive oxygen species (ROS) patterns in MDA-MB-231 cells, but no improvement in ROS was detected in T47D cells. PDBNP caused a change (depolarization) in mitochondrial membrane permeability as well as a cell cycle arrest in the sub-G1 phase, whilst Annexin V/PI assay, followed by confocal microscopy, revealed that MDA-MB-231 cells died due to necrosis.

Paternaet al. (2015) synthesized derivatives from monoterpene-indole alkaloid hydrazone and evaluated their activity against HCT-116 colon cancer. The derivatives containing the 5-bromo-pyridine (dregamine 5-bromo-pyridin-2-ylhydrazone **89** and tabernaemontanine 5-bromo-pyridin-2-ylhydrazone **90**) ring were the most cytotoxic, reducing cell viability by over 95% at 25 µM. The derivative tabernaemontanine phenylhydrazone **91** reduced cell viability to the same level at 50 µM.

Similarly, a Chinese research group synthesized phenanthroindolizidine, which was characterized as 2-(((3,6,7-trimethoxyphenanthren-9-yl)methyl)amino)ethanol **92** and 2-(((2,3,6,7-tetramethoxyphenanthren-9-yl)methyl)amino)ethanol **93**. These compounds exhibited potent effects against the colon cancer cell line HT-29 [113].

## 6. Effect of Alkaloids on Cell Cycle Arrest and Other Anticancer Pathways

The anticancer effect of alkaloids and the mechanistic pathways involved in the anticancer effect of alkaloids are schematically represented in Figure 2. Piperine **51**, an alkaloid commonly isolated from black pepper [98], mediated its anticancer effect via modulation of G1-phase cell cycle arrest and by downstream regulation of cyclins D1 and D3, accompanied by their activating cofactors, cyclin-dependent kinases 4 and 6. Piperine caused downstream expression of phosphorylation of retinoblastoma protein and upstream modulation of p21/WAF1 and p27/KIP1 expression (Figure 2) [97]. The G1 expression for the initiation of Rb protein phosphorylation shape D-type cyclin complexes with Cdk4 and Cdk6 [115,116].On the other hand, to expresses cell cycle progression, it caused inhibition of cyclin-dependent kinases when this complex attached to p21/WAF1 and p27/KIP1, an inhibitor of cyclin-dependent kinases [117,118].

The African medicinal plant *Tabernaemontana elegans* yielded obasinyl-iboga alkaloids dregamine **52,** and tabernaemontanine **53** exhibited significant antiproliferative activity by modulating its effect through G1-phase arrest in colon cancer cell lines with associated downstream regulation in the S phase [99]. Arun et al. (2017) studied the effect of pyranocarbazole alkaloids, murrayazoline compound **61,** and *O*-methylmurrayamine A compound **62** in different stages of the cell cycle; both caused significant cell arrest at the G2/M checkpoint and promoted sub-diploid population at 5.7 µM and 17.9 µM, respectively [102].

The synthetic phenanthroindolizidine alkaloids (**92** and **93**) showed strong anticancer action by modulating multiple singling pathways. These compounds at a low concentration (5 µg/mL) elicited cell cycle arrest, while targeting the G2/M phase. Moreover, a noticeable reduction in cell distribution was observed at the G0/G1 and S phases. In addition, Min et al. (2012) described the effects of another phenanthroindolizidine alkaloid, antofine **41,** in HCT-116 cells, causing cell cycle arrest at the G0/G1 phases and inhibiting the expression of cyclin D1, cyclin E, CDK4, and the transcriptional activity of β-catenin/Tcf. Moreover, this alkaloid also potentiated tumor necrosis factor-a (TNF-α)-induced apoptosis and reduced the expression level of β-catenin and cyclin D1 in SW-480 cells [119].

Mechanistic studies on compounds **6** and **7** showed inhibition of the G0/G1 phase after 24 h exposure [83], with directed cell death absent mitochondrial alterations [120]. The antiproliferative effects of alkaloids **42**–**44** (Table 1) from *Zanthoxylum capense* in HCT-116 cell were shown to be mediated by apoptosis (Figure 1) [94].

Yang et al. (2010) showed that the arrest of the G1 phase in colon cancer cells and the induction of apoptosis are associated with the suppression of nuclear factor-kappaB (NF-kB) activation. The alkaloid also inhibited the phosphorylation/degradation of IkBa and the phosphorylation/translocation of p65. Furthermore, dauricine down-regulated the expression of various NF-kB-regulated genes, including genes involved in cell proliferation (COX-2, cyclin D1, and c-Myc), invasion (MMP-9 and ICAM-1), antiapoptosis (Bcl-2, XIAP, IAP1, and survivin), and angiogenesis (VEGF). The diversity of anticancer mechanisms was also noted with isostrychnopentamine (ISP) **2**, an indolomonoterpenic alkaloid that causes cell cycle arrest in the G2-M phase, and induces apoptosis by the translocation of phosphatidylserine from the inner layer to the outer layer of the plasma membrane, as well as by chromatin condensation, DNA fragmentation, and activation of the caspases 3 and 9 [91].

An experimental finding has illustrated a strong cytotoxic effect of oxymatrine **63**, an alkaloid isolated from *Sophora flavescens* in human colon cancer cells. The underlying mechanism was interfering with the overexpression of human telomerase reverse transcriptase and causing up-regulation of *hTERT*, *c-myc*, *p53*, and *mad1* [121]. The overall effect was concentration dependent (Figure 1) [122].

## 7. Effects on Chemoresistance

Cancer chemotherapy is limited by cellular drug resistance [123,124,125,126]. In this regard, different mechanisms have been implied for this resistance, including cell cycle arrest and repair [127], apoptosis [128], cancer stem cells [129], drug properties/nature [130], and metabolism of the drug [131]. Cell cycle arrest plays a crucial role in drug resistance to chemotherapy in various cancers [127]. Several studies have confirmed the involvement of checkpoint kinase (CHK1) in chemoresistance and even resistance in radiotherapy [132,133,134]. In the case of DNA damage, the complex cellular signaling network becomes activated and induces cell cycle arrest, thus facilitating DNA-repairing events or, in cases of extensive damage, such events trigger apoptosis [135,136,137]

Damage of the cellular DNA of any origin triggers the activation/upstream regulation of ATM(ataxia-telangiectasia mutated),and ATR(ATM- and Rad3-related) kinases coupled with the DNA-dependent protein kinase catalytic subunit [135,138]. These kinases lead to the translocation of components to the sites of damaged DNA and induce cell cycle progression by modulating effector kinases and checkpoint kinase like CHK1, and CHK2 [139]. The activation of checkpoints controlled by ATM/ATR–CHK1/CHK2 stalls cell cycle progression in the G1, S, or G2 phase [140]. G1 arrest is modulated by p53 through p21CIP1/WAF1 up-regulation [141], while in case of DNA damage, apoptosis charged over the situation [130]. In the course of events, p53 loses its regulatory action; as a result, the chemotherapy-induced DNA damage is unable to arrest the cancer cells in the G1 phase and thus trigger apoptosis. Indeed, to arrest the cell cycle in such cancer cells, following the genotoxic exposure, the S and G2/M checkpoints are required to induce DNA repair before entry into mitosis (M phase). Different polysulfanes have shown cell cycle arrest in colorectal cancer [142,143,144].

## 8. Conclusions and Future Prospects

Based on increased cases of cancer and its status as a leading cause of disease worldwide, new effective treatments are required. Among the most common cancer treatments, chemotherapy is an essential tool in getting cancer under control and reducing or treating advanced or aggressive cancers. As noted in the current review, several alkaloids show anti-colon-cancer properties, being able to be cytotoxic against human colon cancer cells. The most exciting substances seem to be those able to act at the level of the checkpoints present at the G1/S and G2/M transitions, which account for the proper replication and division of DNA. These alkaloids hold great potential as novel therapeutic agents for the effective management of colon cancer.

## Figures and Tables

**Figure 1 molecules-27-00920-f001:**
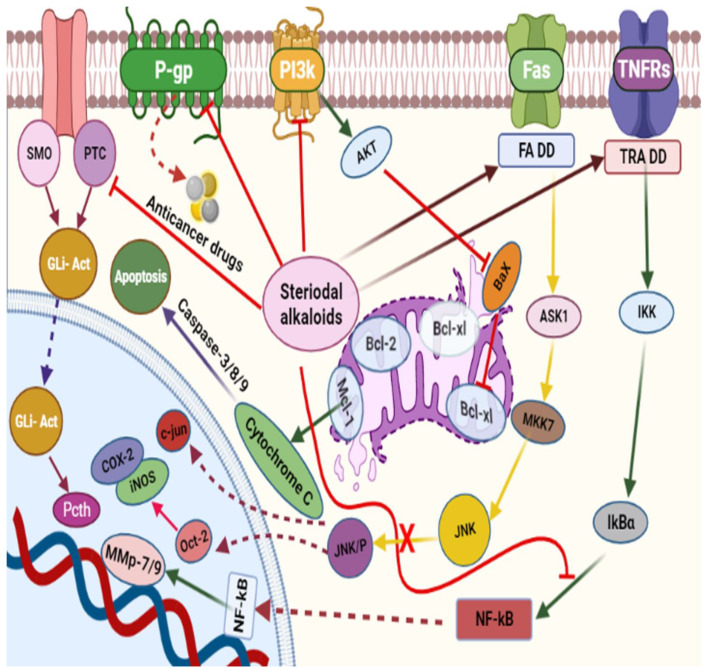
Schematic representation of molecular mechanisms of action involved in anticancer effect of steroidal alkaloids in colonic cancer. Steroidal alkaloids inhibit different pathways such aspg-p, PI3k, Fas, and TNRfs, which are, in turn, linked with different mechanistic pathways exhibiting anticancer effects. Inhibition of Bax and Bcl2 leads to activation of caspase-3,8, and 9 pathways, thus leading to apoptosis.

**Figure 2 molecules-27-00920-f002:**
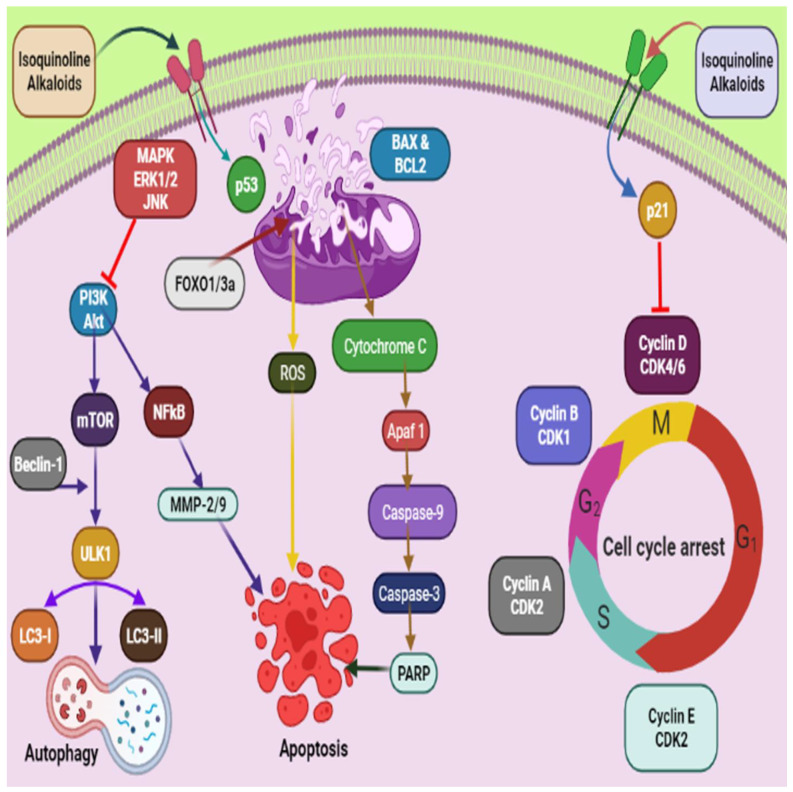
Schematic representation of mechanistic pathways involved in the anticancer effect of alkaloids. Isoquinoline alkaloids exhibit anticancer effect via activating caspase pathway leading to apopstosis. P53 activates different pathways, such as activating ROS, Caspase and inhibiting PI3k. Isoquinoline alkaloids also inhibit Cyclin D CDK 4/6 and causes cell cycle arrest. mTOR pathway is also activated, which leads to autophagy.

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
