# Peer review of "Alkaloids and Colon Cancer: Molecular Mechanisms and Therapeutic Implications for Cell Cycle Arrest"

_molecules, 2022, doi:10.3390/molecules27030920_

Round 1
Reviewer 1 Report
Herbal isolates have shown tremendous pharmacological activities and thus are in use worldwide as medicine. The drug discovery from herbal medicines is to isolate active ingredient from the respective herbs. Authors reported interesting manuscript on alkaloids from herbs possesses anticancer activity against colon cancer.
- Authors reported 2017 cancer cases of US in Introduction. Authors are suggested to update the data of 2020 or 2021.
- Authors are suggested to correct space error throughout the manuscript.
- Authors must add one para about the synthetic colon cancer drugs used in colon cancer with drawbacks.
- Authors have not reported the plant parts (i.e. fruit, stem, leaves, roots etc.) from which isolate obtained.
- Maybe the authors can focus on compound classes of each alkaloid isolated compounds under that.
- Is there any side effect of these isolated compounds? If yes than authors should also discuss them.
- Authors only reported one marine alkaloid nortopsentin. Is there any other marine alkaloid reported for colon cancer? Cite them also.
- Also better to include toxicity studies if available.
- Is there any nano-formulations for these isolated alkaloids? Authors can also add a page for these nanoformulations.
- Figure 1 has no significant explanatory. Why not it should be deleted?
- In Table 1: Better to add the therapeutic dose of each drug used for in vivo activity.
- Give the information about the use of synthetic drug in combination with herbal drug combination formulation used in colon cancer.
- Authors have cited only one reference of year 2020 and no reference of year 2019 and 2021. Without citing new references manuscript cannot be consider for publication.
Author Response
Reviewer 1:
Many thanks to the editor and reviewer for valuable suggestions. We have addressed all of them one by one and highlighted. I am sure their incorporation has greatly aid to the overall strength of our article. The article is revised by an American coauthor, Prof. Dr. Michael Aschner.
Herbal isolates have shown tremendous pharmacological activities and thus are in use worldwide as medicine. The drug discovery from herbal medicines is to isolate active ingredient from the respective herbs. Authors reported interesting manuscript on alkaloids from herbs possesses anticancer activity against colon cancer.
- Authors reported 2017 cancer cases of US in Introduction. Authors are suggested to update the data of 2020 or 2021.
Response: The needful changes have been done.
- Authors are suggested to correct space error throughout the manuscript.
Response: The needful changes have been done.
- Authors must add one para about the synthetic colon cancer drugs used in colon cancer with drawbacks.
Response: The needful changes have been done.
- Authors have not reported the plant parts (i.e. fruit, stem, leaves, roots etc.) from which isolate obtained.
Response: The needful changes have been done.
- Maybe the authors can focus on compound classes of each alkaloid isolated compounds under that.
Response: The needful changes have been done
- Is there any side effect of these isolated compounds? If yes than authors should also discuss them.
Response: The needful changes have been done.
- Authors only reported one marine alkaloid nortopsentin. Is there any other marine alkaloid reported for colon cancer? Cite them also.
Response: The needful changes have been done
- Also better to include toxicity studies if available.
Response: The needful changes have been done.
- Is there any nano-formulations for these isolated alkaloids? Authors can also add a page for these nanoformulations.
Response: Needful content has been added and highlighted.
- Figure 1 has no significant explanatory. Why not it should be deleted?
Response: The needful changes have been done.
- In Table 1: Better to add the therapeutic dose of each drug used for in vivo activity.
Response: The needful changes have been done
- Give the information about the use of the synthetic drug in combination with herbal drug combination formulation used in colon cancer.
Response: Needful content has been added and highlighted.
- Authors have cited only one reference of year 2020 and no reference of year 2019 and 2021. Without citing new references manuscript cannot be consider for publication.
Response: The needful changes have been done.
Reviewer 2 Report
Comments:
The manuscript reviewed the “Alkaloids and Colon Cancer: Molecular Mechanism and Therapeutic Implications Via Cell Cycle Arrest”. The manuscript is beneficial to the readership of journal. However, the manuscript should be improved thoroughly. There are too many technical mistakes.
Specific comments:
In formatting, please follow strictly to the journal’s guidelines. Different font size was detected ie. in the affiliation and text, Citation in text ie. [6-8], the use of semicolon and comma, spacing between words, units ml or mL and etc. Authors are advised to thoroughly checked the manuscript for guidelines adherence.
Line 28-29: Please quote the latest figures from WHO
Line 29: Please quote the latest figures
Line 30: spacing between “was” and 600,920.
Line 52: small letters for Angiotensin Converting Enzyme
Line 78: The abbreviations G1, S, G2, M should have introduced earlier when they first mentioned in text.
Line 109-126: No reference citation in number
Line 115: African
Line 121: Since chaetominine already been designated as 3, thus, it should be used as compound 3. The same for the rest.
Line 128: Solanumaculeastrum? Spacing between words is a major problem in the manuscript.
Line 132: Sanguinaria canadensis
Line 144-148: No reference cited in number? Please check thoroughly in the whole manuscript.
Line 149: E. rutaecarpa, please give full name.
Line 151-164: Spacing
Line 169: Please be consistent in the format for citation in text. It should be using number in bracket, [xx].
Line 170: Zanthoxylumcapense?
Line 171: Guava ViaCount is a trademark.
Line 176: gramichunosin
Line 188: melodinine V
Line 189: Melodinushenryi?
Line 193: Argemonemexicana?
Line 195: compounds 57 and 59
Line 197: Murrayakoenigii?
Line 202: font size?
Line 220: Reference?
Line 221-223: Reconstruct the sentence
Line 226: analogue 66
Line 229: Reference format?
Line 237: Reference format?
Line 242: 5-phenyl…
Line 243-252: compounds’ names are in small letters.
Line 253: Berberine, should number as 80 before line 255.
Line 278-283: Reconstruct the sentence, and including the reference.
Line 299: Compound 51
Line 309: yieldedvobasinyl-iboga?
Line 310: compounds 52 and 53
Line 313: compounds 61 and 62
Line 323: alkaloid 42
Line 342: compound 23
Line 350: compound 2
Line 355: Figure 2 has already mentioned in line 305.
Line 361: oxymatrine, compound numbering?
Line 362: Sophoraflavescens?
Line 364: font size?
Line 367-368: Reconstruct the sentence
Line 406: Table 1 is not mentioned in text.
Line 407: Table 2 is not mentioned in text.
Authors should individually discuss each cell cycle in the sub-sections. It would be better if they are accompanied by schematic diagram.
There are many abbreviations in the manuscript. Authors are advised to incorporate List of Abbreviations.
Authors are strongly advised to re-organise Table 1 and 2.
Reference should be in number enclosed with bracket.
Please standardise the chemical structures, they are not consistent. Some structures are not clear.
Please indicate compound’s name with numbering.
Please carefully check the plant species names, bold is not needed.
Authors should create a new column indicating the potent IC50.
References: Please follow strictly to the journal’s requirement. Inconsistencies in reference format were found.
Author Response
Reviewer 2
Many thanks to the editor and reviewer for valuable suggestions. We have addressed all of them one by one and highlighted. I am sure their incorporation has greatly aid to the overall strength of our article. The article is revised by an American coauthor, Prof. Dr. Michael Aschner.
Comments:
The manuscript reviewed the “Alkaloids and Colon Cancer: Molecular Mechanism and Therapeutic Implications Via Cell Cycle Arrest”. The manuscript is beneficial to the readership of journal. However, the manuscript should be improved thoroughly. There are too many technical mistakes.
Specific comments:
In formatting, please follow strictly to the journal’s guidelines. Different font size was detected ie. in the affiliation and text, Citation in text ie. [6-8], the use of semicolon and comma, spacing between words, units ml or mL and etc. Authors are advised to thoroughly checked the manuscript for guidelines adherence.
Line 28-29: Please quote the latest figures from WHO
Line 29: Please quote the latest figures
Line 30: spacing between “was” and 600,920.
Line 52: small letters for Angiotensin Converting Enzyme
Line 78: The abbreviations G1, S, G2, M should have introduced earlier when they first mentioned in text.
Line 109-126: No reference citation in number
Line 115: African
Line 121: Since chaetominine already been designated as 3, thus, it should be used as compound 3. The same for the rest.
Line 128: Solanumaculeastrum? Spacing between words is a major problem in the manuscript.
Line 132: Sanguinaria canadensis
Line 144-148: No reference cited in number? Please check thoroughly in the whole manuscript.
Line 149: E. rutaecarpa, please give full name.
Line 151-164: Spacing
Line 169: Please be consistent in the format for citation in text. It should be using number in bracket, [xx].
Line 170: Zanthoxylumcapense?
Line 171: Guava ViaCount is a trademark.
Line 176: gramichunosin
Line 188: melodinine V
Line 189: Melodinushenryi?
Line 193: Argemonemexicana?
Line 195: compounds 57 and 59
Line 197: Murrayakoenigii?
Line 202: font size?
Line 220: Reference?
Line 221-223: Reconstruct the sentence
Line 226: analogue 66
Line 229: Reference format?
Line 237: Reference format?
Line 242: 5-phenyl…
Line 243-252: compounds’ names are in small letters.
Line 253: Berberine, should number as 80 before line 255.
Line 278-283: Reconstruct the sentence, and including the reference.
Line 299: Compound 51
Line 309: yieldedvobasinyl-iboga?
Line 310: compounds 52 and 53
Line 313: compounds 61 and 62
Line 323: alkaloid 42
Line 342: compound 23
Line 350: compound 2
Line 355: Figure 2 has already mentioned in line 305.
Line 361: oxymatrine, compound numbering?
Line 362: Sophoraflavescens?
Line 364: font size?
Line 367-368: Reconstruct the sentence
Line 406: Table 1 is not mentioned in text.
Line 407: Table 2 is not mentioned in text.
Authors should individually discuss each cell cycle in the sub-sections. It would be better if they are accompanied by schematic diagram.
There are many abbreviations in the manuscript. Authors are advised to incorporate List of Abbreviations.
Authors are strongly advised to re-organise Table 1 and 2.
Reference should be in number enclosed with bracket.
Please Standardise the chemical structures, they are not consistent. Some structures are not clear.
Response: The needful changes have been done.
Please indicate compound’s name with numbering.
Please carefully check the plant species names, bold is not needed.
Authors should create a new column indicating the potent IC50.
References: Please follow strictly to the journal’s requirement. Inconsistencies in reference format were found.
Response: Thank You very much for your kind suggestions and valuable comments, the needful changes have been done and highlighted.
Reviewer 3 Report
In general, throughout the manuscript, the authors do not use recent citations (the most recent citation appears to be from 2018). In particular, in the introduction, to describe the current epidemiological situation of colon cancer, the authors took as reference the articles of 2017, but the Globocan statistics are different, albeit slightly.
Especially in consideration of the fact that this type of manuscript would be a review, my advice is to review the bibliography, updating it.
Author Response
Reviewer 3:
Many thanks to the editor and reviewer for valuable suggestions. We have addressed all of them one by one and highlighted. I am sure their incorporation has greatly aid to the overall strength of our article. The article is revised by an American coauthor, Prof. Dr. Michael Aschner.
In general, throughout the manuscript, the authors do not use recent citations (the most recent citation appears to be from 2018). In particular, in the introduction, to describe the current epidemiological situation of colon cancer, the authors took as reference the articles of 2017, but the Globocan statistics are different, albeit slightly.
Especially in consideration of the fact that this type of manuscript would be a review, my advice is to review the bibliography, updating it.
Response: The needful suggested changes have been done in the manuscript and highlighted.
Round 2
Reviewer 2 Report
In the revised version, some comments but not limited to:
Line 34: 600,920,
Line 124-140: Reference numbers?
Line 143: It should be Sanguinaria canadensis
Line 147: compounds’ numbers are in “bold”. Please check and revise accordingly.
Line 184: gramichunosin [small letter G], chemical names are in small letters unless starting in new sentence
Line 196: melodinine V
Line 216-223: Reference numbers?
Line 226: (Table 2)
Line 224-231: Reference in numbers?
Line 231-232: Plakinamines N (1) and O (2), two additional steroidal alkaloids from Corticium niger, as well as two established plakinamine compounds, were discovered (3, 4).
Question 1: Compound number (1) and (2) have been designated to evodiamine 1 and Isostrychnopentamine (ISP) 2. Plakinamines N (1) and O (2) shouldn't be numbered as 1 and 2.
Question 2: What does it mean by “were discovered (3, 4)”?
Line 248-258: Reference numbers?
Line 260-292: Reference numbers?
Line 293-298: Reference numbers?
Line 314: It should be dregamine 52
Line 317: It should be compound 61 and compound 62
Line 331: Table 1
Line 333-344: Reference numbers?
Line 350: It should be oxymatrine 63
Authors are advised to carefully cite the references in text as numbers enclosed with a bracket.
Chemical structures are not consistent, as mentioned in the first revision. Obviously, structure of compound 8 is larger than some other structures. Authors should use the same formatting to draw all the structures.
Table 1 and Table 2 should have another column to indicate the IC50 or EC50, as mentioned in the first revision. Tables are not organised. For instances, references are shown as (Ogasawara et al., 2001) as well as [83], IC50 should not be shown as bold, EC50 (50 should be subscript). Plant species names should not be shown as bold. Some mistakes: Sanguinaria Canadensis, it should be Sanguinaria canadensis, Menispermum (why underlined), Evodiarutaecarpa?? Zanthoxylumcapense?? Tabernaemontanacorymbosa?? vobasinyl−iboga Alkaloids shouldn’t be mentioned. Argemonemexicana?? Murrayakoenigii??
Authors are advised to carefully revise and check the manuscript before submission.
Author Response
Dear Editor
Many thanks to the editor/reviewer for valuable inputs. We have addressed all of them one-by-one.
Regards

Reviewer 3 Report
The authors took into consideration the comments and suggestions of the reviewers.
Author Response
Dear Editor
Many thanks to the editor/reviewer for acknowledgment of our revision.
Regards
